# Analysis of Crude, Diverse, and Multiple Advanced Glycation End-Product Patterns May Be Important and Beneficial

**DOI:** 10.3390/metabo14010003

**Published:** 2023-12-19

**Authors:** Takanobu Takata, Togen Masauji, Yoshiharu Motoo

**Affiliations:** 1Division of Molecular and Genetic Biology, Department of Life Science, Medical Research Institute, Kanazawa Medical University, Uchinada 920-0293, Ishikawa, Japan; 2Department of Pharmacy, Kanazawa Medical University Hospital, Uchinada 920-0293, Ishikawa, Japan; masauji@kanazawa-med.ac.jp; 3Department of Internal Medicine, Fukui Saiseikai Hospital, Wadanakacho 918-8503, Fukui, Japan

**Keywords:** lifestyle-related disease (LSRD), advanced glycation end-products (AGEs), crude AGE pattern, diverse AGE pattern, multiple AGE pattern, gas chromatography-mass spectrometry (GC-MS), matrix-assisted laser desorption-mass spectrometry (MALDI-MS), electrospray ionization-mass spectrometry (ESI-MS), Kampo medicines

## Abstract

Lifestyle-related diseases (LSRDs), such as diabetes mellitus, cardiovascular disease, and nonalcoholic steatohepatitis, are a global crisis. Advanced glycation end-products (AGEs) have been extensively researched because they trigger or promote LSRDs. Recently, techniques such as fluorimetry, immunostaining, Western blotting, slot blotting, enzyme-linked immunosorbent assay, gas chromatography-mass spectrometry, matrix-assisted laser desorption-mass spectrometry (MALDI-MS), and electrospray ionization-mass spectrometry (ESI-MS) have helped prove the existence of intra/extracellular AGEs and revealed novel AGE structures and their modifications against peptide sequences. Therefore, we propose modifications to the existing categorization of AGEs, which was based on the original compounds identified by researchers in the 20th century. In this investigation, we introduce the (i) crude, (ii) diverse, and (iii) multiple AGE patterns. The crude AGE pattern is based on the fact that one type of saccharide or its metabolites or derivatives can generate various AGEs. Diverse and multiple AGE patterns were introduced based on the possibility of combining various AGE structures and proteins and were proven through mass analysis technologies such as MALDI-MS and ESI-MS. Kampo medicines are typically used to treat LSRDs. Because various compounds are contained in Kampo medicines and metabolized to exert effects on various organs or tissues, they may be suitable against various AGEs.

## 1. Introduction

Lifestyle-related diseases (LSRDs), including diabetes mellitus (DM), cardiovascular disease (CVD), and nonalcoholic steatohepatitis (NASH), are a global crisis, mostly in developing countries [1,2,3,4,5]. This is attributed to people consuming excess saccharides (e.g., glucose and fructose), proteins, and lipids. However, LSRDs were also known in ancient times, when only a few affluent individuals, such as those belonging to royalty or nobility, could afford to consume them excessively because saccharide-sweetened ingredients such as *Gynostemma pentaphyllum* (Thunb.) (Japanese name: amachazuru; Chinese name: jaogulan) were used in daily life [6]. Moreover, ancient Chinese medicine considered the concept of DM symptoms [7,8]. Advanced glycation end products (AGEs), which are generated from saccharides/their metabolites or derivatives and proteins, are associated with LSRDs [1,2,3,4,5,9]. Although AGEs have been investigated since the early 20th century, when the Maillard reaction was proven [9], their relationship with human health dates back to ancient times. AGEs have been analyzed using techniques such as fluorimetry [10], immunostaining [2,11,12], Western blotting [2,13], slot blotting [2,14,15,16], enzyme-linked immunosorbent assay (ELISA) [11,17,18], gas chromatography-mass spectrometry (GC-MS) [19], matrix-assisted laser desorption-mass spectrometry (MALDI-MS) [20], and electrospray ionization-mass spectrometry (ESI-MS) [21,22]. Researchers have investigated LSRDs and AGEs using these technologies. AGEs can be classified as intra- and extracellular AGEs [1,2]. Intracellular AGEs are generated in cells and can induce cell dysfunction and death [2,9,10,11,12,13,14]. Extracellular AGEs include (i) AGEs in body fluids (e.g., blood [2,10,11,14,17,18,23], saliva [24], and urine [17,19]) and (ii) dietary AGEs [9,25]. Dietary AGEs are generated in beverages and foods that are manufactured or cooked by heating. Because AGEs in the blood, saliva, and urine are released or leaked from organs, some researchers have investigated their potential as biomarkers against disease or organ dysfunction [11,17,18,19,23,24]. Extracellular AGEs can bind to receptors for AGEs (RAGE) and toll-like receptor 4 (TLR4) and induce cytotoxicity, including inflammation [26,27]. Researchers can analyze the molecular weights of AGEs, their structure, and the glycated amino acids in peptide sequences using the aforementioned technologies. In this article, we categorized the approaches for analyzing AGEs as (i) crude, (ii) diverse (types 1 and 2), and (iii) multiple AGEs (types 1 and 2) patterns. In the crude AGE pattern, one type of saccharide or its metabolites or derivatives generate certain AGE structures in cells [2,28]. In the type 1 diverse AGE pattern, some AGE structures can be modified against one type of protein [29,30,31], whereas in type 2, one type of AGE structure can be modified using certain proteins [32]. In the type 1 multiple AGE pattern, some AGE structures are modified by one protein molecule [31], whereas in type 2, one AGE structure is modified by more than two proteins via intramolecular covalent bonds [2,31]. Using this approach towards the relationships between AGE structure and proteins, the diversity and multiplicity of AGEs can be explored, and new targets of investigation can be discovered for researchers who aim to find cures or prevent AGE-associated diseases.

Kampo medicines are traditional medicines that were introduced from China but developed uniquely in Japan from the fifth to the nineteenth centuries [33,34,35]. Modern Kampo medicines are produced from extracts by pharmaceutical companies governed by several national laws in Japan since approximately 1950 [35,36]. The Japan Society for Oriental Medicine has published “Standards of Reporting Kampo Products” to provide researchers with information on Kampo products in English [37]. Kampo medicines have been subjected to randomized controlled trials to obtain clinical evidence [38,39]. For Japanese industry, academia, and government, promoting “Kampo medicines for cancer supportive care” is one of the goals for the future of Kampo medicines [40,41,42]. Despite numerous efforts, the components of Kampo medicines have not yet been identified [43,44,45]. These medicines generate various metabolites that affect certain organs [42,43,44,45]. Therefore, Kampo medicines may inhibit the generation of intracellular AGEs, suppress AGE-RAGE/TLR4 signaling, and cure and prevent LSRD triggered or promoted by AGEs.

## 2. Techniques for Analyzing AGEs

### 2.1. Fluorimetry

AGEs fluoresce, and measuring this phenomenon is very simple and useful because it enables researchers to detect and understand AGE structure [10]. However, other fluorescent substances may also be considered.

### 2.2. Immunostaining

The benefit of immunostaining is that AGE accumulation in cells or tissues can be visualized [2,11,12,46]. Moreover, AGE-positive areas can be enumerated. Kehm et al. measured arg-pyrimidine- and pentosidine-positive areas in the pancreatic islets (AGEs area/pancreatic islets) [11], while van Heijist et al. revealed that heat shock protein (HSP) 27 and arg-pyrimidine were co-located in the immunostaining images of squamous cell carcinoma tissue in patients with non-small cell lung cancer; arg-pyrimidine-modified HSP27 was also located in this tissue [46]. Considering the characteristics that require anti-AGE antibodies, immunostaining [2,11,12,46], Western blotting [2,13,32], slot blotting [2,14,15,16], and ELISA [10,17,18] can be grouped together. Moreover, if the structure of AGEs remains unclear (e.g., AGEs with an unknown structure generated from methylglyoxal and bovine serum albumin [BSA] [47]), it can still be examined using anti-AGE antibodies produced against the AGEs for the antigen. However, this method has drawbacks. Ikeda et al. researched the monoclonal and polyclonal anti-*N*^ε^-carboxymethyl-lysine (CML) antibodies [48,49]. They revealed that some anti-CML antibodies could probe non-CML epitopes in proteins. Therefore, researchers may not completely assess the regulation of AGEs with only anti-AGE antibodies, which causes non-specificity in the results obtained from non-mass spectrometry analyses such as immunostaining, Western blotting, slot botting, and ELISA. In these analyses, researchers may require a control that has the epitope without the targeted AGEs and can be probed using anti-AGEs antibodies.

### 2.3. Western Blotting

In Western blot analysis, AGE-modified proteins on the membrane are detected by chemiluminescence or fluorescence, and the molecular weight of each AGE-modified protein is visualized [2,13,32]. Previously, we reported one type of glyceraldehyde-derived AGEs (GA-AGEs), which some researchers refer to as toxic AGEs (TAGE), in a human pancreatic ductal cell line (PANC-1) treated with glyceraldehyde [2]. The structure of TAGE was shown to contain a 1,4-dihydropyrazine ring [5]; however, it has not been proven definitively. TAGE-modified proteins have been detected using polyclonal antibodies [2]. Mastrocola et al. fed high-fat high-sugar to C57 and OB/OB mice and quantified CML- and *N*^ε^-carboxyethyl-lysine (CEL)-modified proteins in murine skeletal muscle based on their detected bands [32].

### 2.4. Slot Blotting

Slot blot analysis is generally performed to quantify AGEs [2,14,15,16]. In 2017, we developed a novel slot-blot method with the following characteristics: (i) a lysis buffer containing tris-(hydroxymethyl)-aminomethane (Tris), urea, thiourea, and 3-[3-(cholamidopropyl)-dimethylammonio]-1-propanesulfonate (CHAPS), and (ii) polyvinylidene fluoride (PVDF) membranes [2]. We considered that this lysis buffer could promote the ability of samples containing AGE-modified proteins to suitably probe the PVDF membrane [16]. We analyzed TAGE and 1,5-anhydro-D-frucotose-derived AGEs (1,5-AF-AGEs) in cells and tissues using TAGE-modified BSA and 1,5-AF-AGEs-modified BSA as standards [2,14,15,16,50]. Bronowica-Szyełko et al. analyzed serum methylglyoxal-derived AGEs (MGO-AGEs) using slot blotting [17]. Although they selected a PVDF membrane, no reagents were added to the serum sample. Dot blot analysis is similar to slot blot analysis, differing only in the dilatation of the area on the membrane [51].

### 2.5. ELISA

The concept of ELISA is similar to that of slot blotting, i.e., samples are probed into 96-well microplates, and AGEs are identified and quantified using anti-AGE antibodies [11,17,18,23,24,52]. Samples, such as plasma [11], serum [17,23], saliva [24], urine [18], and tissue lysates [52], have been analyzed. Ruiz-Meana et al. quantified CML-modified proteins in human cardiac tissue using CML-modified BSA as a standard [52]. Although many targets in the samples were AGE-modified proteins, Kashiwabara et al. quantified a free-type of pentosidine in urine, which is a unique challenge considering the difficulty of producing antibodies against low molecular weight compounds [18].

### 2.6. GC-MS

GC-MS has been used to perform absolute quantification of free-type AGEs [19,53,54,55]. However, AGE-modified proteins cannot be detected and analyzed because the conditions of GC-MS analysis (i) include a sample molecular weight range of 100–1000 Da, (ii) samples must be modified to their ester derivatives, (iii) samples should be heated and degraded at 250–500 °C, and (iv) the molecular weight of the target must be proven [19,56]. Therefore, AGE-modified proteins must be hydrolyzed to obtain free-type AGEs and undergo esterification to obtain free AGE-ester derivatives (Figure 1) [19].

Baskai et al. performed the absolute quantification of CML and CEL using stable isotope-substituted CML and CEL as internal standards. The CML and CEL in the samples could be quantified because their retention times were the same and the mass ion peaks (*m*/*z*) differed [19,53,54,55].

### 2.7. MALDI-MS and ESI-MS

MALDI-MS and ESI-MS have promoted the investigation of proteomics since approximately 1990, and these instruments can be connected to liquid chromatography (LC) to produce LC-MALDI-MS and LC-ESI-MS [57,58,59]. Researchers have widely used LC-ESI-MS because of the ease of sample injection. Based on the peptide sequence database, peptide ion peaks and their fragment ion peaks have been calculated, and proteins have been identified [57,58,59], contributing to the investigation of AGEs. MALDI-MS/ESI-MS and nuclear magnetic resonance (NMR) have been used to identify the structure of free-type AGEs, such as 3-deoxyglucosoine-derived lysine dimer (1,3-di(*N*^ε^-lysino)-4-2-(2,3,4-trihydroxybutyl)-imidazoium salt) (DOLD), glyoxal-lysine dimer (1,3-di(*N*^ε^-lysino) imidazolium salt) (GOLD), methylglyoxal-lysine dimer (1,3-di(*N*^ε^-lysino)-4-methyl-imidazolium salt) (MOLD), vesperlysne A, glucosepane, tetrahydropyrimidine, glycolaldehyde-pyridine, and crossline [60,61,62]. Although novel free-type AGEs have been identified by analyzing individual ion peak data using MALDI-MS/ESI-MS and NMR spectrum data, the structure, ion peak pattern, and degradation of free-type AGEs have been analyzed and their data inputted into databases using only MALDI-MS/ESI-MS. Based on the development of technology for free-type AGE analysis, free-type AGEs have been quantified in various samples [21,63,64,65]. However, when AGE-modified proteins in samples are subjected to acid hydrolysis (e.g., hydrochloric acid), free-type AGEs can be obtained for quantification (Figure 2) [21,63,64,65].

Analyzing AGE-modified peptides is challenging because typical databases do not contain glycation information, although information on methylation and acetylation is available [57,58,59]. When researchers analyze proteins, peptides can be identified using databases that can predict the possibility of methylation and acetylation. Therefore, numerous peptides and proteins can be identified based on peptide information. Although information on glycosylation is normally unavailable, this modification can be removed from peptide sequences during ionization and does not inhibit peptide identification [59]. In contrast, glycation cannot be removed from peptides during the ionization step of MALDI-MS/ESI-MS analysis, and its information is unavailable; therefore, AGE-modified peptides cannot be identified (Figure 3) [57,58,59]. However, many researchers have attempted to analyze AGE-modified proteins and succeeded in identifying amino acids (e.g., lysine and arginine) in AGE-modified proteins by comparing the ion peaks of normal and glycated peptides [22,30,31,51,66,67,68]. If researchers can input information on AGE modification against peptides in databases, AGE-modified proteins can be efficiently identified (Figure 3) [28].

## 3. Intra-/Extracellular AGEs and LSRDs

### 3.1. Intracellular AGEs and LSRDs

Previously, we reported that PANC-1 cells and a human pancreatic islet β cell line 1.4E7 underwent TAGE-induced cell death, and microtubule-associated protein light chain 3 (LC)-I, LC3-II, and p62 in 1.4E7 cells were downregulated [2,14]. In addition, Suh et al. reported that methylglyoxal treatment of a murine pancreatic islet β cell line (RIN-m5F) generated MGO-AGEs, induced cell death, and reduced insulin secretion [47]. Ohno et al. demonstrated that glucoselysine and CML accumulate in the eye lenses of diabetic Wistar rats [21], and Bellier et al. reported that MGO-AGEs induce DM and cancer [22]. Furthermore, Kehm et al. revealed that a carbohydrate-free, high-fat diet and carbohydrate-rich diet (CRD) fed to New Zealand obese (MZO) mice (obese, diabetes-prone model mice) resulted in the considerable accumulation of argpyrimidine and pentosidine in the pancreatic islets [11]. Ruiz-Meana et al. found that CML-modified proteins accumulate in murine cardiac tissue and promote senescence in vivo [52], and we reported that intracellular TAGE may induce the death of normal human cardiac fibroblasts in vitro [15]. Mastrocola et al. reported that CML- and CEL-modified proteins accumulated in the skeletal muscle of OB/OB mice fed a high-fat, high-sugar diet, and their AGEs might be associated with myosteatosis [32]. We showed that generating 1,5-AF-AGEs induces cell death in a human hepatic cell line (HepG2) [50]. Based on these investigations, we hypothesized that intracellular AGEs may be associated with LSRDs, including DM, cancer, CVD, sarcopenia, and NASH.

### 3.2. Extracellular AGEs and LSRDs

#### 3.2.1. AGEs in Fluids and LSRDs

Intracellular AGEs are generated in various cells and released or leaked into fluids (e.g., blood, saliva, and urine) as extracellular AGEs [10,11,17,18,19,23,24]. These AGEs can induce cytotoxicity, thereby including inflammation, because RAGE and TRL4 are expressed on the surfaces of various cells [1,11,26]. Furthermore, researchers have attempted to show that AGEs in the fluid correlate with the onset/progression of specific diseases, proving that AGEs could be biomarkers for these diseases [11,18,19,23,24,69,70,71]. The most beneficial and authentic biomarkers are (i) generated in specific organs and (ii) detected upon disease onset/progression, or organ dysfunction [72,73,74,75]. Alanine aminotransferase (ALT) and aspartate aminotransferase (AST) are beneficial biomarkers of liver dysfunction, including nonalcoholic fatty liver (NAFL) and NASH, because high levels of ALT and AST are produced in the liver, unlike in other organs [72,73]. Creatinine kinase (CK) is a biomarker of heart dysfunction, for example, CVD [74,75]. CK in the blood increases when cardiomyocytes die due to CVD or is released from skeletal muscle during strenuous exercise. Although many AGEs can be generated/accumulated in certain organs, AGE has not been conclusively proven as a disease biomarker. However, Kehm et al. reported that plasma CML increased in MZO mice fed a CRD [11], and Kuang et al. reported increased plasma CML in patients with DM [69]. Notably, Kato et al. simultaneously analyzed four AGEs [CML, CEL, *N*^δ^-(5-hydro-5-methyl-4-imidazolone-2-yl)-ornithine (5-hydroxy-5-methylimidazolone) (MG-H1), and *N*^ω^-carboxymethyl-arginine (CMA)] in the serum of nephropathy patients using LC-ESI-MS and showed that MG-H1 dramatically increased relative to the other three AGEs [70]. Lirwinowicz et al. determined the structure of a synthetic, melibiose-derived AGE (MAGE) and showed that MAGE in plasma correlated with NASH in a clinical study [71]. Although the structure of TAGE remains unclear, serum TAGE levels are associated with DM, CVD, NASH, infertility, cancer, and Alzheimer’s disease [5] and may be rare biomarkers with utility in determining the risk of various LSRDs.

#### 3.2.2. Dietary AGEs and LSRDs

AGEs are produced in beverages and foods because of the presence of saccharides (e.g., glucose and fructose) and proteins, and heating treatment during manufacturing or cooking induces the Maillard reaction [1,25,76,77]. Because they can combine RAGE and TRL4 [78], AGEs may induce cytotoxicity via the dietary AGEs-RAGE/TLR4 axis. Based on the “Takayama Study”, Wada et al. suggested that CML, a dietary AGE, was significantly associated with an increased risk of male cancer [25]. Chen et al. focused on three AGEs, CML, CEL, and MG-H1, which are harmful to human health (e.g., oxidative stress and inflammation in the gut) [76]. Notwithstanding, we have focused on the relationship between dietary AGEs and oral, esophageal, and gastric epithelial cells [1]. RAGE is expressed on oral [79,80], esophageal [81], and gastric epithelial cells [82], and TLR4 is expressed on oral [83], esophageal [84], and gastric epithelial cells [85]. Because these epithelial cell types are exposed to air and direct physical contact with dietary AGEs, the dietary AGEs-RAGE/TRL4 axis may appear.

## 4. Categories of Free-Type AGEs Based on Original Saccharides and Their Metabolites/Derivatives

Until the beginning of the 21st century, seven categories of free-type AGEs were established: glucose-derived AGEs (Glc-AGEs, AGE-1), GA-AGEs (AGE-2), glycolaldehyde-derived AGEs (AGE-3), MGO-AGEs (AGE-4), glyoxal-derived AGEs (GO-AGEs, AGE-5), 3-deoxyglucosone-derived AGEs (3DG-AGEs, AGE-6), and acetaldehyde-derived AGEs (AA-AGEs) [1,5,9,86]. These groups were named based on the original saccharides and their metabolites/derivatives, which originated from the free-type AGE structure. However, subsequent research has changed the categorization of AGEs, owing to the development of various technologies for analyzing AGE structures [1,19,20,21,22,60,61,62]. Some reports indicate that glyceraldehyde-derived pyridinium (GLAP) [87,88,89], trihydroxy-triosidine [90], lys-hydroxy-triosidine [90], arg-hydroxy-triosidine [90], triosidine-carbaladehyde [90], MG-H1 [91], and argpyrimidine [92] have been generated from glyceraldehyde in vitro (categorized as GA-AGEs), although MG-H1 and argpyrimidine have been categorized as MGO-AGEs [93,94,95,96]. In contrast, Basakal et al. used GC-MS to reveal that CML, which had not been categorized as an MGO-AGEs, was generated from methylglyoxal in vitro [55]. Wang et al. reported that CML-modified proteins in a rat cardiomyocyte cell line (H9c2) treated with methylglyoxal increased, as shown by Western blotting [97]. Litwinowicz et al. classified the MAGE that they synthesized as AGE-10 [71]. The categories of MGO-AGEs and GA-AGEs may change in the future (Figure 4).

## 5. Crude AGE Pattern

We focused on the generation and accumulation of some types of AGEs from one type of saccharide metabolite/derivative (e.g., glyceraldehyde) in cells (Figure 5). We named this phenomenon the crude AGE pattern. Various AGEs that originate from the same compound may induce cytotoxicity and cell death. Previously, we reported that TAGE is generated and accumulated in PANC-1 cells treated with 1, 2, and 4 mM glyceraldehyde for 24 h via cell immunostaining, Western blotting, and slot blotting with a polyclonal anti-TAGE antibody [2]. However, we did not investigate the possibility that other types of AGEs may have been generated. Accordingly, Senavirathana et al. treated PANC-1 cells with 1, 2, and 4 mM glyceraldehyde for 48 h under conditions similar to those of our study and revealed that the cells generated and accumulated GLAP, argpyrimidine, and MG-H1 (Figure 5) [28]. Their analysis identified and quantified GLAP-, MG-H1-, and argpyrimidine-modified proteins by ESI-MS but not by Western or slot blotting with antibodies. Furthermore, they succeeded in proving that glyceraldehyde could produce three types of AGEs, although they did not investigate the possibility that TAGE might be generated. We believe that the study by Senavirathana et al. provides important data verifying the crude AGE pattern [28]. Notably, they showed that the ratio of GLAP, MG-H1, and argpyrimidine differed at different concentrations, although the incubation time for the glyceraldehyde treatment of PANC-1 was the same. These phenomena may be important because they reveal the conditions or pathways for generating each AGE structure. Furthermore, GLAP, MG-H1, and argpyrimidine might be associated with cytotoxicity and cell death, though we considered that only TAGE induced these phenomena [2,28]. However, we might have to focus on the possibility that glyceraldehyde generated various types of AGEs and induced death/dysfunction in cells.

## 6. Diverse AGE Pattern

### 6.1. Type 1 Diverse AGE Pattern

The type I diverse AGE pattern concerns AGE structures that can be modified in one type of protein (but one molecule of protein) (Figure 6). Although this pattern should be predictable based on the crude AGE pattern, this is difficult to prove owing to insufficient technology.

Norkin et al. treated recombinant human HSP90 with methylglyoxal and performed immunoprecipitation during production, which was analyzed by Western blotting using anti-MG-H1 and anti-argpyrimidine antibodies, with positive bands being detected [29]. Although this study suggested that MG-H1 and argpyrimidine were modified with some of the same peptide sequences as HSP90, we were unable to prove the type 1 diverse AGE pattern. Researchers cannot prove this pattern using antibodies alone, and mass spectrometry is required. Compared with only free-type AGEs, which are able to be analyzed with GC-MS [19,53,54,55], MALDI-MS and ESI-MS are suitable for analyzing AGE-modified proteins [21,30,31,63,64,65]. Although MALDI-MS/ESI-MS equipment and their databases are appropriate to prove the type 1 diverse AGE pattern, they must detect the appropriate AGE-modified peptides, which is difficult. Oya-Ito et al. successfully detected an appropriate AGE-modified peptide using LC-MALDI-MS [31]. They treated recombinant human HSP27 and phosphorylated HSP27 (p-HSP27) with methylglyoxal; their analysis revealed 22 and 23 AGE-modified peptides in HSP27 and p-HSP27, respectively. Furthermore, they revealed that some AGE types (e.g., HG-M1 and argpyrimidine) were modified at the same amino acid residue in the same sequence [31]. These data prove the type I diverse AGE pattern; we have shown some of the data in Figure 7. Two types of “HGYISRCFTR (131–140)” in HSP27, HG-M1 or argpyrimidine, were detected (Figure 7a). Although R (135) and R (139) were present in this sequence, HG-M1 and argpyrimidine were only modified at R (135). In “SRAQJGGPEAAR (187–198)”, R (188) was modified with MG-H1 or arg-pyrimidine (Figure 7b). Based on the investigation by Oya-Ito et al., we consider that the type I diverse AGE pattern can be proven.

### 6.2. Type 2 Diverse AGE Pattern

The type II diverse AGE pattern refers to a type of AGE structure that can be modified by some types of proteins (Figure 8). This has generally been detected using Western blotting through sodium dodecyl sulfate (SDS) polyacrylamide gel electrophoresis (PAGE) (SDS-PAGE) [2,32,97]. This pattern was easily proven by examination using an anti-AGE antibody because the molecular weight and isoelectric point of each AGE protein were different. In contrast, Papadatki et al. [97] reported methylglyoxal modification of actin and myosin using LC-ESI-MS.

## 7. Multiple AGE Pattern

### 7.1. Type 1 Multiple AGE Parttern

We named the type I multiple AGE pattern because some types of AGE structures are modified by one protein molecule (but not one type of protein) (Figure 9). Although this pattern should be predictable based on the crude and type I diverse AGE patterns, analyzing this problem using more than one type I diverse AGE pattern is challenging. When AGE-modified proteins are used for tryptic diagnosis and their peptides are analyzed using MALDI-MS or ESI-MS, the peptides modified with more than two AGE structures should be detected (Figure 9).

Although this condition is challenging to achieve, Oya-Ito et al. used MALDI-MS to obtain two MG-H1 modified peptides, “SPAVAAPAYSRALSRQJSSGVSE (65–87)” and “IRHTADRWR (88–96)” in recombinant human HSP27 treated with methylglyoxal (Figure 10) [31]. Their data were suitable and sufficient to prove the type I multiple AGE pattern. Two MG-H1s modified R (75) and R (79) in the same peptide, and two MG-H1s modified R (89) and R (94) in HSP27.

### 7.2. Type 2 Multiple AGE Pattern

The type II multiple AGE pattern involves modification of the AGE structure in more than two proteins via intermolecular covalent bonds (Figure 11). The AGE structure, which can combine more than two amino acid residues in the protein (e.g., TAGE [5], DOLD [60], GOLD [60], MOLD [60], glucosepane [60], vesperlysine A [60], crossline [60], and triosidines [90]), can generate a complex of more than two proteins. This protein complex is produced via covalent bonds without disulfide bonds, and their covalent bonds cannot be destroyed when analyzed by Western blotting using SDS-PAGE. We reported that PANC-1 cells were treated with glyceraldehyde, and the high molecular bands of HSP27, HSP70, and HSP90β were detected with Western blot analysis using SDS-PAGE [2]. These data suggested the possibility that the structure of GA-AGEs generated the homo- or heterodimer of HSPs, though we were unable to prove the real chimerical structure of GA-AGEs. GLAP [87], MG-H1 [91], and argpyrimidine [92] can combine one amino acid residue but not more than two. Therefore, they could not generate the complex of proteins, though they were generated by glyceraldehyde in PANC-1 cells [28]. In contrast, TAGE [5] and some types of triosidines [90] were generated from glyceraldehyde and can combine more than two amino acid residues. They might generate a complex of proteins.

The AGE structure produces both intra- and intermolecular covalent bonds (Figure 12). Therefore, one type of AGE structure generates a complex of more than two proteins, while the other type may be modified via intramolecular covalent bonds (Figure 12). Nevertheless, researchers cannot prove that only one type of AGE structure generates a complex of some proteins (combining more than two proteins) with only anti-AGE antibodies because anti-AGE antibodies cannot recognize the difference between intra- and intermolecular covalent bonds in the AGE structure. Although the anti-D2 antibody recognized the “D2 type of AGEs structure and amino acids (4 and 13)” in protein A, the D1 type AGEs structure contributed to the intermolecular covalent bond between protein A and B. Therefore, while the anti-D2 antibody recognized this complex, we cannot state conclusively that D2-type AGEs contributed to producing the complex of proteins A and B (Figure 12).

MALDI-MS or ESI-MS analyses cannot accurately identify the one type of AGE structure that combines more than two proteins. Peptide sequences without AGE modification were identified based on the normal database using MALDI-MS/ESI-MS analysis (Figure 13). However, the AGE structure of the complex of peptides originating from more than two proteins could not be identified, although the molecular weight of the AGE modification (e.g., D1 in Figure 13) was automatically inputted into the database. If AGE structures combine with one protein residue, such as CML, CEL, or MG-H1, the molecular weight and mass ion peak of the peptide modified with these AGE structures can be calculated (Figure 6, Figure 7, Figure 9, and Figure 10). However, this technology can be employed only if one type of protein is targeted for analysis. Because we could not predict the molecular weight of the complex of the AGE structure and peptides with more than two origins, the calculated molecular weight (and *m*/*z*) could not identify the amino acid sequences and AGE structure (Figure 13).

## 8. Investigation of AGEs and LSRDs Based on the Three Patterns

Current technologies, including manufacturing anti-AGE antibodies, GC-MS, MALDI-MS, and ESI-MS, can analyze the structure of AGEs (free types of AGEs), their modified amino acid residues, and proteins that are modified with the AGE structure. Based on the development of an analysis method for AGEs, we described the crude, diverse, and multiple AGE patterns. We believe that these patterns can focus on the relationship between AGEs and LSRDs (e.g., DM, CVD, NASH, infertility, cancer, and Alzheimer’s disease) at the macro- to micro-levels. Although simple examination systems should be constructed for basic investigation, several researchers have made novel discoveries, and the range of AGEs available can be expanded. Therefore, investigating the relationship between AGEs and LSRDs may provide new research directions. The crude AGE pattern was proven in cellulosis and in vitro (Figure 5) [2,28]. If the regulation of one type of AGE was associated with the death/dysfunction of cells, other types of AGEs, including those that have been analyzed, might induce these phenomena. In contrast, one type of AGEs might induce the death/dysfunction of cells by cooperating with other types of AGEs. Multiple studies that many researchers have performed can confirm the link between individual AGEs and LSRDs, if such a correlation exists. This possibility applies to the diverse and multiple AGE patterns. If researchers observe that an AGE structure is modified in an individual protein, revealing its dysfunction, another type of AGE structure may be modified against it to inhibit its activity (Figure 6 and Figure 9). If one type of AGE-modified protein showed dysfunction or harmed the cell, other proteins modified with the same type of AGE structure did not show abnormal activity (Figure 8). In contrast, one type of AGE structure may promote dysfunction or harm the cell by cooperating with other types of AGE structures, whereas some types of structure may trigger/promote LSRDs simultaneously. The most difficult issue to approach is the type II multiple AGE pattern (Figure 11, Figure 12 and Figure 13). The investigation based on the type II multiple AGE pattern remains inconclusive. These issues may apply to both intracellular and extracellular AGEs. On the AGEs-RAGE/TRL4 axis, the effects of various AGEs may be detected. When researchers investigate the treatment of LSRDs triggered or promoted by various extracellular, body fluid, or dietary AGEs based on crude, diverse, and multiple AGE patterns, analyzing their mechanisms is difficult. However, differentiating between these patterns may be important and beneficial for researchers.

## 9. Effects of Kampo Medicines on Intra- and Extracellular AGEs

Researchers investigating the treatment and prevention of LSRDs promoted or triggered by various AGEs do not agree conclusively on the medicines to be employed against various AGEs. However, we believe that Kampo medicines containing crude drugs may be beneficial against various AGEs. Modern extracts of Kampo medicines are manufactured, and the quality control of some beneficial components of Kampo medicines is ensured by several national laws in Japan (Figure 14) [35,36]. The numerous compounds present in Kampo medicines cannot be detected, even using three-dimensional HPLC analysis [98,99,100]. Various compounds are ingested, digested in the stomach, absorbed into the small intestine, and metabolized in the liver [1]. These processes produce metabolites that are transported to organs or tissues, such as the mouth, small and large intestines, brain, and skeletal muscle, and they can undergo further metabolism and non-enzymatic reactions (Figure 15) [43,44]. Although all compounds in Kampo medicines have not been identified, the utility of Kampo medicines in treating diseases or inducing beneficial effects against syndromes has been investigated [42,45]. Kampo medicines have been used to treat LSRDs such as DM [101,102] and NASH [103,104]. Yamakawa et al. reported that Bofutstushosan (Chinese name: Fang-Feng-Tong-Sheng-San) contained various low-molecular compounds and insist it may be useful for the treatment of obesity [105]. Uneda et al. revealed that Bofutsushosan significantly reduced body mass index (BMI) in a clinical study [106]. Because BMI was reduced, we consider that Bofutsushosan was able to improve obesity. Although Seitai (Chinese name: Qing-Dai) was known as the crude drug that had therapeutic effects for ulcerative colitis, which may be associated with LSRDs, it improved the pulmonary arterial hypertension of the patients with ulcerative colitis [107,108]. Kampo medicines may inhibit the generation of intracellular AGEs, block the extracellular AGEs-RAGE/TRL4 axis, and suppress phenotypes via AGEs-RAGE/TRL4 axis signaling. Ańanco et al. [109] and Yadav et al. [110] focused on low molecular weight compounds that are present in natural products (e.g., catechin [109], epicatechin [109], epicatechin gallate [109], gallic acid [110], and gallotannin [110]) as they might inhibit AGEs generation and suppress AGEs-RAGE/TRL4 axis signaling. In contrast, Zhi et al. revealed that Jin-Si-Wei (Japanese name: Shikunshito), which consists of *Ginseng*, *Atracylodos macrocephala*, *Poria cocos*, and *Glycyrrhiza uralensis*, decreased amyloid β_1–42_, amyloid precursor protein, and RAGE in the brains of APP^swe^/PS1^ΔE9^ transgenic mice [111]. Because Jin-Si-Wei reduced RAGE, it may suppress AGEs-RAGE axis signaling and improve Alzheimer’s disease. Liu et al. also reported the potential of the decoction of *Angelica sinensis*, *Zingiberis Rhizoma Recens*, and *Mutton* (Japanese name: Toukisyoukyouyounikuto. Chinese name: Dang-Gui-Sheng-Jiang-Yang-Rou-Tang) to inhibit pro-inflammatory factors through AGEs-RAGE axis signaling in diarrhea-predominant irritable bowel syndrome [112]. Although Toukisyoukyouyounikuto is not contained in modern Kampo medicines because they were not described in the current Japanese Pharmacopoeia by several national laws in Japan [33,34,35,36], they may be focused on the effects of inhibition of AGEs-RAGE axis signaling in the future. The potential of some Kampo medicines to suppress AGEs-RAGE axis signaling has been revealed, though researchers have not made the mechanisms sufficiently clear because Kampo medicines contain various compounds that would be metabolized and treated with non-enzymic reactions (Figure 15) [43,44]. On the contrary, the mechanisms by which each Kampo medicine or Crude drug suppresses the generation of individual AGEs (e.g., CML, CEL, and CMA) remain unclear. However, Ban et al. reported that the aqueous extract of *Trapa biopiosa* Roxb., a crude drug, inhibited MG-H1 generation from ribose-derived gelatin in vitro [113]. Tominaga et al. reported that the aqueous methanol extract of *Drosera tokainsis* inhibits the generation of CML and CMA from ribose-derived gelatin in vitro [114]. Although Kampo medicines should generally be administered orally and their compounds are metabolized to exert effects in the human body, previous research provides important background [43,44,45,115]. Studies in which the characteristics of Kampo medicines are reproduced must clarify their effects against intra- and extracellular AGEs in vitro and in vivo [1]. Although the mechanisms underlying the various effects of Kampo medicines have not been sufficiently elucidated, they may be beneficial against various types of AGEs because each compound and its derivatives can act against various types of AGE structures.

## 10. Conclusions

In the present review, we introduced the crude, diverse, and multiple AGE patterns. We believe that these patterns suggest the need to perform more complex analyses to reveal the structure of AGEs, AGE modifications in peptide sequences, and novel perspectives on the investigation and treatment of LSRDs associated with AGEs. Furthermore, we believe that Kampo medicines may be beneficial against various AGEs that trigger or promote LSRDs. Therefore, analyzing these patterns of AGEs may be important and beneficial.

## Figures and Tables

**Figure 1 metabolites-14-00003-f001:**
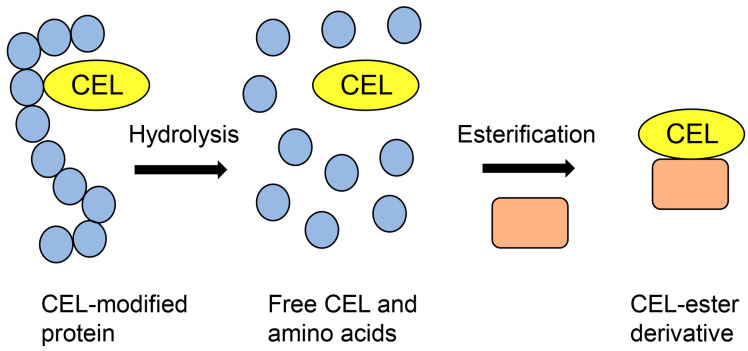
Model image of the preparation of a CEL-ester derivative for GC-MS analysis. Closed blue circles represent amino acids. Closed peach squares represent compounds that have hydroxyl groups. CEL: *N*^ε^-carboxyethyl-lysine.

**Figure 2 metabolites-14-00003-f002:**
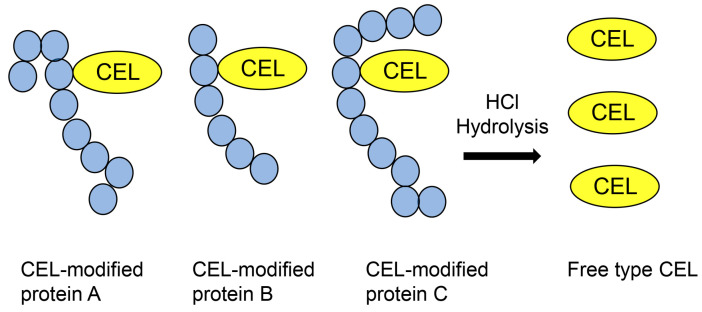
Hydrochloride hydrolysis of various CEL-modified proteins and production of free CEL for analysis using MALDI-MS or ESI-MS. CEL: *N*^ε^-carboxyethyl-lysine; MALDI-MS, matrix-assisted laser desorption-mass spectrometry; ESI-MS, electrospray ionization-mass spectrometry.

**Figure 3 metabolites-14-00003-f003:**
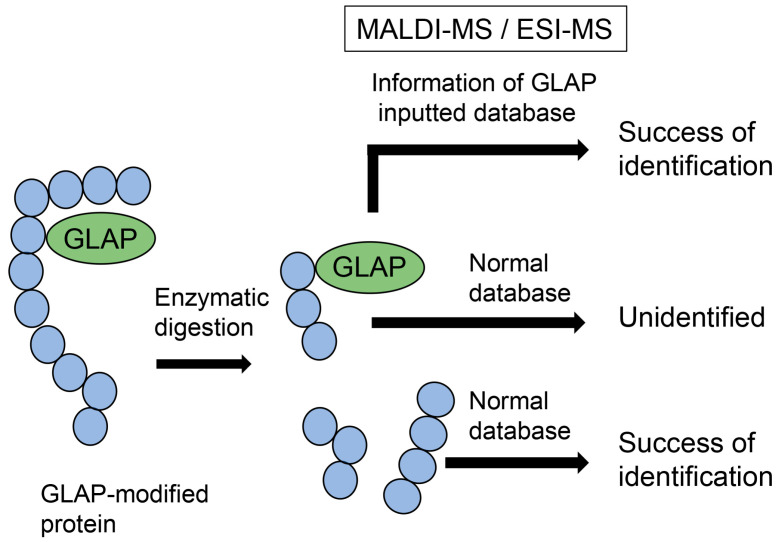
Model image of MALDI-MS or ESI-MS analysis of GLAP-modified protein. GLAP, glyceraldehyde-derived pyridinium; MALDI-MS, matrix-assisted laser desorption-mass spectrometry; ESI-MS, electrospray ionization-mass spectrometry.

**Figure 4 metabolites-14-00003-f004:**
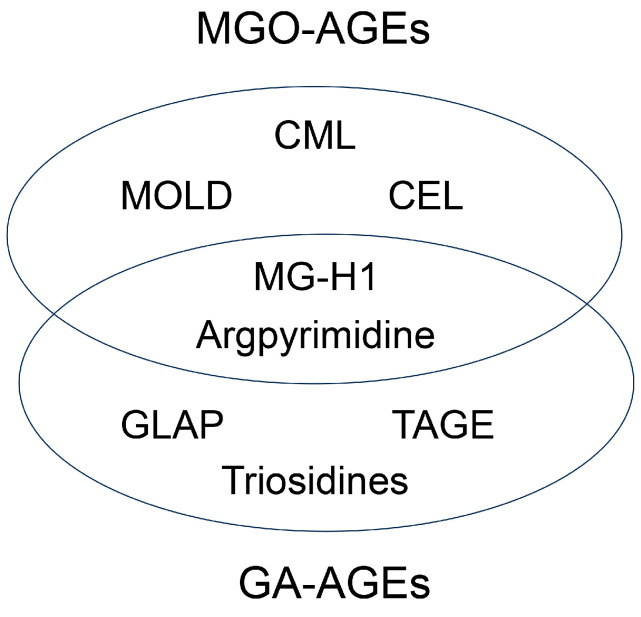
New model of categorizing MGO-AGEs and GA-AGEs. Numbers indicate references. MGO-AGEs, methylglyoxal-derived AGEs; GA-AGEs, glyceraldehyde-derived AGEs; CML, *N*^ε^-carboxymethyl-lysine [55,97]; CEL, *N*^ε^-carboxyethyl-lysine [55,96]; MOLD, methylglyoxal-lysine dimer (1,3-di(*N*^ε^-lysino)-4-methyl-imidazolium salt) [5,60]; MG-H1, *N*^ε^-(5-hydro-5-methyl-4-imidazolone-2-yl)-ornithine [91,93,94]; Argpyrimidine [92,95,96]; GLAP, glyceraldehyde-derived pyridinium [87,88,89]; TAGE, toxic AGEs [2,5]; Triosidines [90].

**Figure 5 metabolites-14-00003-f005:**
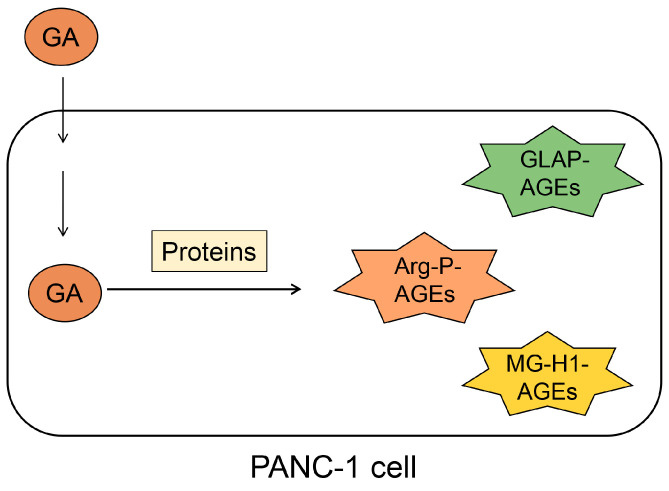
Illustration of the generation and accumulation of various AGEs in PANC-1 cells treated with glyceraldehyde [28] via the crude AGE pattern. GA, glyceraldehyde; Arg-P, argpyrimidine; MG-H1, *N*^δ^-(5-hydro-5-methyl-4-imidazolone-2-yl)-ornithine. GLAP, glyceraldehyde-derived pyridinium; AGEs, advanced glycation end-products.

**Figure 6 metabolites-14-00003-f006:**
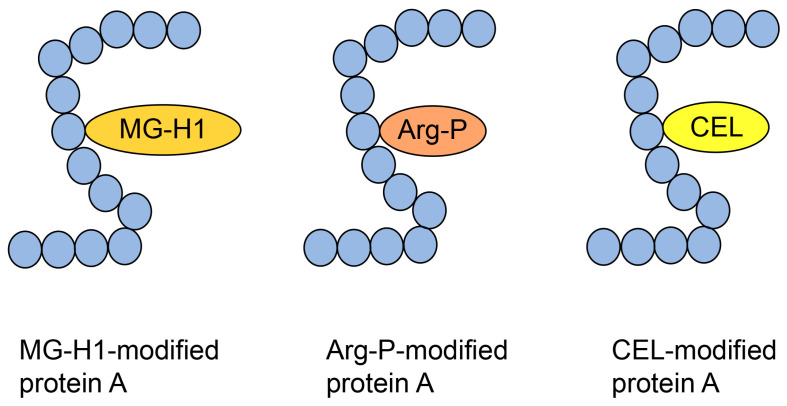
Type 1 diverse AGE pattern. Each protein A can be modified by MG-H1, Arg-P, and CEL. Blue closed circles indicate the amino acids. MG-H1, *N*^δ^-(5-hydro-5-methyl-4-imidazolone-2-yl)-ornithine; Arg-P: argpyrimidine; CEL, *N*^ε^-carboxyethyl-lysine.

**Figure 7 metabolites-14-00003-f007:**
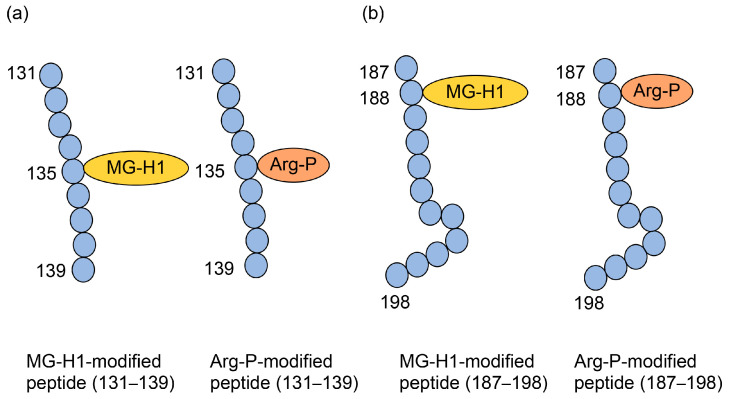
MG-H1 and argypyrimidine-modified peptides in HSP27 [31]. Closed blue circles represent amino acids. Numbers indicate amino acids in the HSP27 sequence. MG-H1, *N*^δ^-(5-hydro-5-methyl-4-imidazolone-2-yl)-ornithine; Arg-P: argpyrimidine; (**a**) The amino acid sequence: HGYISRCFTR (131–140). (**b**) Amino acid sequence: SRAQJGGPEAAR (187–198).

**Figure 8 metabolites-14-00003-f008:**
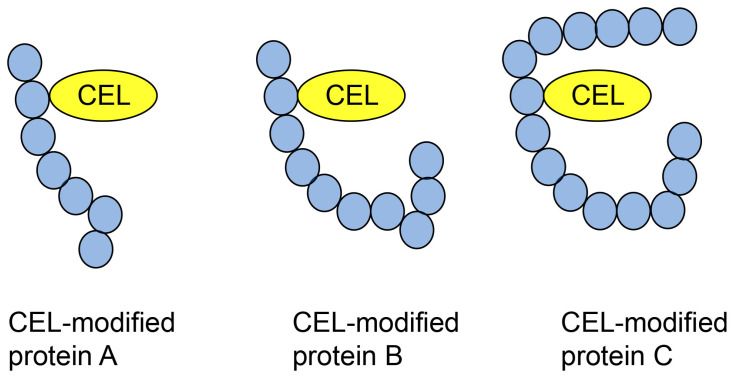
Type 2 diverse AGE pattern. CEL-modified proteins A–C [32]. Closed blue circles represent amino acids. CEL, *N*^ε^-carboxyethyl-lysine.

**Figure 9 metabolites-14-00003-f009:**
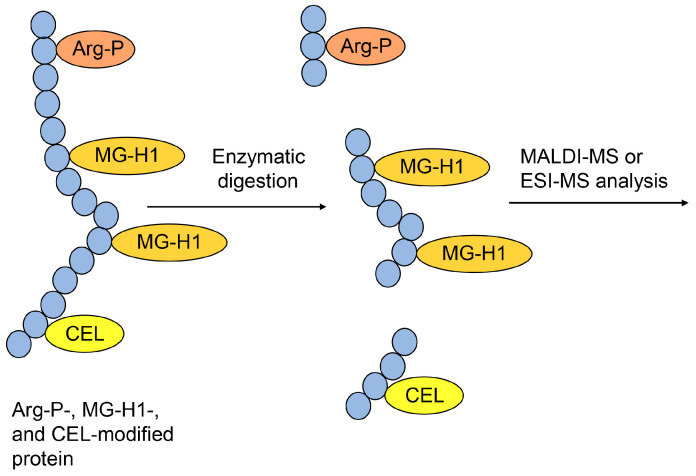
Model of the type I multiple AGE pattern and the MALDI-MS or ESI-MS analysis of Arg-P-, MG-H1-, and CEL-modified proteins. The peptide modified with two MG-H1s verified the type I multiple AGE pattern. Closed blue circles represent amino acids. Arg-P: argpyrimidine; MG-H1, *N*^δ^-(5-hydro-5-methyl-4-imidazolone-2-yl)-ornithine; CEL, *N*^ε^-carboxyethyl-lysine.

**Figure 10 metabolites-14-00003-f010:**
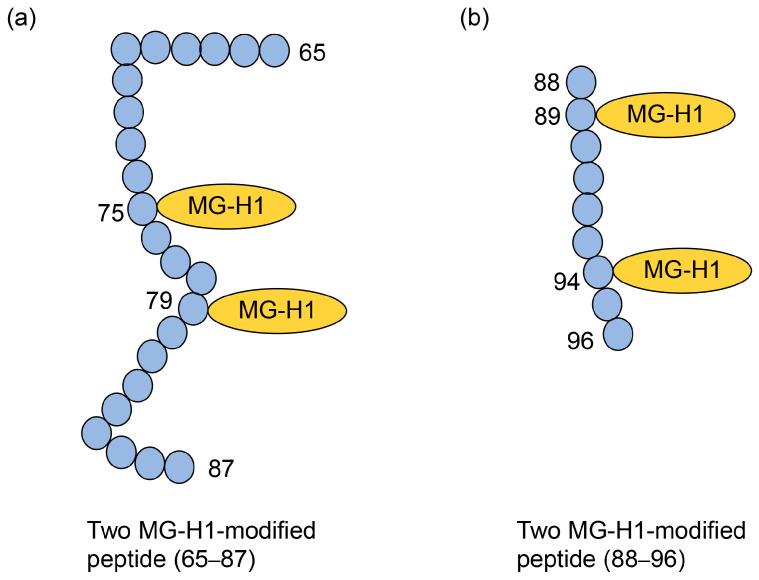
Two MG-H1-modified peptides (65–87 and 88–96) in recombinant human HSP27 [31]. Closed blue circles represent amino acids. Numbers indicate amino acids in the HSP27 sequence. MG-H1, *N*^δ^-(5-hydro-5-methyl-4-imidazolone-2-yl)-ornithine. (**a**) The amino acid sequence: SPAVAAPAYSRALSRQJSSGVSE (65–87). (**b**) The amino acid sequence: IRHTADRWR (88–96).

**Figure 11 metabolites-14-00003-f011:**
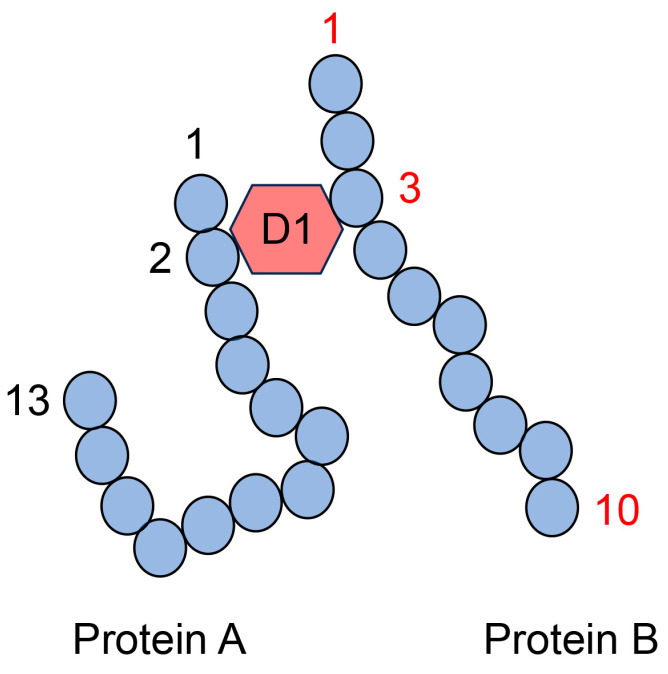
Model image of the type II multiple AGE pattern shows only an intermolecular covalent bond. D1: AGE structure that can combine between the second amino acid residue in protein A and the third amino acid residue in protein B. Closed blue circles represent amino acids; black and red numbers represent the number of amino acid residues in proteins A and B, respectively.

**Figure 12 metabolites-14-00003-f012:**
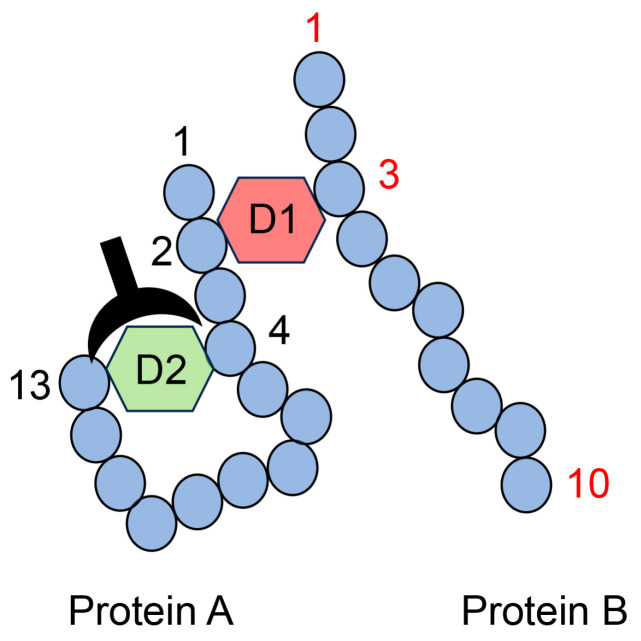
Model image of the type II multiple AGE pattern shows both inter- and intramolecular covalent bonds. D1: AGE structure combining the second amino acid residue in protein A and the third amino acid residue in protein B. D2: AGE structure combining the fourth and thirteenth amino acids in protein A. D1 and D2 are different AGE structures. A closed black plow represents an anti-D2-antibody. Closed blue circles represent amino acids; black and red numbers represent the number of amino acid residues in proteins A and B, respectively.

**Figure 13 metabolites-14-00003-f013:**
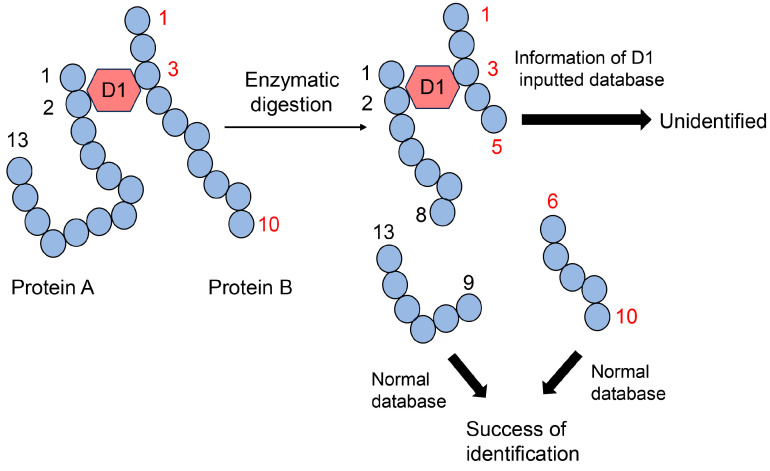
MALDI-MS or ESI-MS analysis of the protein A and B complex, combined via D1. D1: AGE structure that can combine the second amino acid residue in protein A and the third amino acid residue in protein B. Closed blue circles indicate amino acids. Black and red numbers represent the number of amino acid residues in proteins A and B, respectively.

**Figure 14 metabolites-14-00003-f014:**
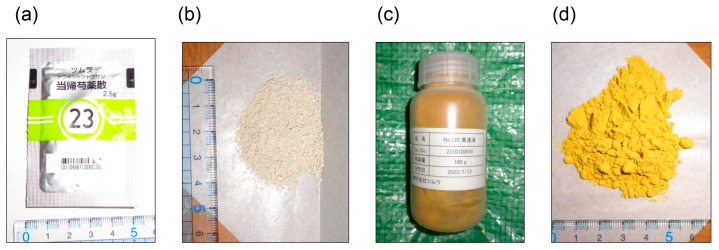
Modern Kampo medicines from Tsumura Co. (Tokyo, Japan) (**a**) Tokishakuyakusan (2.5 g) packaged for use in hospitals or pharmacies. (**b**) Tokishakuyakusan (2.5 g). (**c**) Orento in a bottle, prepared for research at Kanazawa Medical University but not for use in hospitals or pharmacies. (**d**) Orento.

**Figure 15 metabolites-14-00003-f015:**
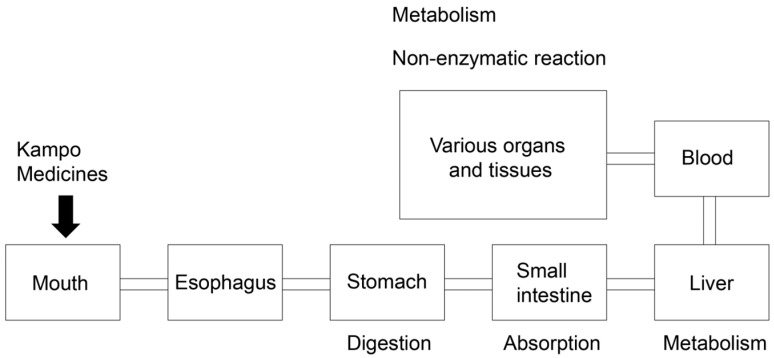
Schematic diagram of oral Kampo medicine administration in the body. Medicines are digested in the stomach, absorbed in the small intestine, metabolized in the liver, passed into the blood, and transported to various organs and tissues.

## Data Availability

The data are contained within the article.

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
