# Peer review of "Analysis of Crude, Diverse, and Multiple Advanced Glycation End-Product Patterns May Be Important and Beneficial"

_metabolites, 2023, doi:10.3390/metabo14010003_

Round 1

Reviewer 1 Report

Comments and Suggestions for Authors

The manuscript submitted by Takata et al., entitled "Crude, diverse, and multiple advanced glycation end products theories and Kampo medicine application based on their characteristic" elegantly described the various tools and methods to assess levels of AGEs, different types of AGE precursors based on their tissue localization in terms of intra- or extra-localization, endogenous sources via diet, and different metabolites of AGE. The authors also discussed in detail different AGE theories citing interesting literature available on different AGE theories and highlighting the possible limitations of previously published studies. The submitted manuscript holds vital and clinically relevant information regarding the role and importance of oxidative stress and reactive oxygen species in different metabolic and cardiovascular diseases.

However, as the title of the manuscript clearly states the application of Kampo medicine to different AGE theories based on their respective characteristics, the authors did not elaborate comprehensively on the specifics of the effect of Kampo medicines against AGE. 90% of the manuscript is focused on AGE characteristics and different AGE theories, whereas only the last section before the conclusion briefly discusses the Kampo medicine's effect against AGE which looks inappropriate as the title suggests that the main goal of the manuscript is to address the role or impact of the Kampo medicine on the AGE.

Either the authors should consider changing the title or should enhance the section discussing the impact of Kampo medicine on AGE.   

Author Response

Response Letter to Reviewers’ Comments

 Responses to Reviewer 1

Dear Reviewer 1:

Thank you for giving us the opportunity to submit a revised draft of our manuscript titled “Analysis of Crude, Diverse, and Multiple Advanced Glycation End-Product Patterns May be Important and Beneficial” to Metabolites (manuscript ID: 2724249). We appreciate the time and effort the reviewers have taken to provide their valuable feedback on our manuscript; their comments have enriched the manuscript and produced a more balanced account of our research. The manuscript has been revised by a professional English editor (Editage) to address all grammatical and syntax errors and improve the overall readability of the document.

We changed the title of this manuscript, and the new title is “Analysis of Crude, Diverse, and Multiple Advanced Glycation End-Product Patterns May be Important and Beneficial.” We have also replaced references 48 and 49 in the first submitted manuscript with the new Ref 48 and 49 (PMID 8672512 and 9744751). More, the new reference (Ref.109) was inserted in Section 9.

The following technical terms were revised:

“Pattern” instead of “Theory”

“Crude AGE pattern” instead of “Crude AGEs theory”

“Diverse AGE pattern” instead of “Diverse AGEs theory”

“Multiple AGE pattern” instead of “Multiple AGEs theory”

We corrected the character in Figure 13 (We removed “?” in Figure 13).

Comment 1: However, as the title of the manuscript clearly states the application of Kampo medicine to different AGE theories based on their respective characteristics, the authors did not elaborate comprehensively on the specifics of the effect of Kampo medicines against AGE. 90% of the manuscript is focused on AGE characteristics and different AGE theories, whereas only the last section before the conclusion briefly discusses the Kampo medicine's effect against AGE which looks inappropriate as the title suggests that the main goal of the manuscript is to address the role or impact of the Kampo medicine on the AGE. Either the authors should consider changing the title or should enhance the section discussing the impact of Kampo medicine on AGE.  

Response 1: Following your suggestions, we change the title of our manuscript. The new title is “Analysis of Crude, Diverse, and Multiple Advanced Glycation End-Product Patterns May be Important and Beneficial.”

Reviewer 2 Report

Comments and Suggestions for Authors

In section 2, the authors provide an overview of the different methods for analysis of AGEs. Maybe the authors could also discuss some (specificity) problems of the non-mass spectrometry methods? (This reviewer remembers older publications, where researchers claimed to have found age-dependent increasing CML levels in mivce, but careful examinations of their western blots strongly suggested increased levels of immunoglobuline in samples); maybe the authors could comment on the specificity of the methods and which controls are required.

The authors introduced three theories: crude, diverse and multiple AGEs theories. For this reviewer it is not clear why the authors called them „theories“ and what is the content of the theories.

For example:

According to the abstract (and section 5.), the crude AGEs theory is „based on the fact“ that a single carbohydrate or its derivative can lead to the formation of various AGEs. That different or various AGEs can be formed from the same carbohydrate or metabolite is thus a fact (and has been documented in many publications). A fact is not a theory. It remains unclear what the content (beyond the facts mentioned) of their theory is.

The type 1 diverse AGEs theory appear to propose that one specific protein can be modified by different AGEs (section 6./ Figure 6). The authors then discuss the paper by Kinoshita et al., who specifically examined CMA modification of collagen and identified CMA pöeptides in glyoxal-modified type III collagen. The authors conclude that „Kinoshita et al. could not completely prove the type 1 diversity AGEs theory“, but that was not the aim of Kinoshita et al. and it is not clear why the experiments in Kinoshita et al. could have the potential to prove the „type 1 diversity AGEs theory“ (in the view of this reviewer the Kinoshita et al. experiments are not able to prove the theory, and were also not designed to do so).

The meaning of many sentences are not clear (likely gramatically not correct; though this reviewer don’t want to judge that, as he is not a native speaker):

Only a few examples:

line 339/340: „Because GC-MS can be used to

line 348: „We introduced one of these CMA-modified peptides (Figure 7); do the authors mean „We show one ......“?

line 351: „(one type of protein, but one molecular)“; meaning of „one molecular“ is unclear – is the expression incomplete?

line 462/463: „The procedure by which one type of AGE structure combines more than two proteins is difficult if researchers perform using MALDI-MS or ESI-MS analyses.“: does it refer to the „procedure“ by which AGEs combines more than two proteins or the procedure researcher are using to exmine this?

line 464: „The peptide sequence without modifying the AGE structure ...“; does it mean „The peptide sequence without AGE-modification“?

Minor points:

It would be helpful for the reader to list all abbreviations that are used more than once in the list at the end of the manuscript (e.g. 1,5-AF-AGEs, MGO-AGEs, TAGE).

Figure 7: the legend states that „One of the CMA-modified peptides ....“ is shown; but actually 3 peptides are shown in the Figure!

Figure 9, caption: should be „CEL-modified“ not „GEL-modified“

Author Response

Response Letter to Reviewers’ Comments

Responses to Reviewer 2

Dear Reviewer 2:

Thank you for giving us the opportunity to submit a revised draft of our manuscript titled “Analysis of Crude, Diverse, and Multiple Advanced Glycation End-Product Patterns May be Important and Beneficial” to Metabolites (manuscript ID: 2724249). We appreciate the time and effort the reviewers have taken to provide their valuable feedback on our manuscript; their comments have enriched the manuscript and produced a more balanced account of our research. The manuscript has been revised by a professional English editor (Editage) to address all grammatical and syntax errors and improve the overall readability of the document.

We changed the title of this manuscript, and the new title is “Analysis of Crude, Diverse, and Multiple Advanced Glycation End-Product Patterns May be Important and Beneficial.” We have also replaced references 48 and 49 in the first submitted manuscript with the new Ref 48 and 49 (PMID 8672512 and 9744751). More, the new reference (Ref.109) was inserted in Section 9.

The following technical terms were revised:

“Pattern” instead of “Theory”

“Crude AGE pattern” instead of “Crude AGEs theory”

“Diverse AGE pattern” instead of “Diverse AGEs theory”

“Multiple AGE pattern” instead of “Multiple AGEs theory”

We corrected the character in Figure 13 (We removed “?” in Figure 13).

Comment 1: In section 2, the authors provide an overview of the different methods for analysis of AGEs. Maybe the authors could also discuss some (specificity) problems of the non-mass spectrometry methods? (This reviewer remembers older publications, where researchers claimed to have found age-dependent increasing CML levels in mivce, but careful examinations of their western blots strongly suggested increased levels of immunoglobuline in samples); maybe the authors could comment on the specificity of the methods and which controls are required.

Response 1: Although we have not provided the information that Reviewer 2 suggested, we understand the problem that anti-AGEs antibodies can probe non-targeted proteins. This specificity problem, which does not occur in the mass spectrometry analysis, can occur in immunostaining, western blot, slot bot, and ELISA.

Ikeda et al. studied the monoclonal and polyclonal anti-CML antibodies (Ref. 48, 49). They revealed some anti-CML antibodies could probe non-CML epitopes in proteins. Therefore, researchers may not be able to completely assess the regulation of AGEs with only anti-AGEs antibody. This is a specificity problem observed in non-mass spectrometry analyses, such as immunostaining, western blot, slot bot, and ELISA. Researchers may need the control, which has the epitope without the targeted AGEs, that can be probed using anti-AGEs antibody in these analyses. We have elaborated on this in Section 2.2. (Yellow highlighted).

Comment 2: The authors introduced three theories: crude, diverse and multiple AGEs theories. For this reviewer it is not clear why the authors called them „ theories“ and what is the content of the theories. For example: According to the abstract (and section 5.), the crude AGEs theory is „based on the fact“ that a single carbohydrate or its derivative can lead to the formation of various AGEs. That different or various AGEs can be formed from the same carbohydrate or metabolite is thus a fact (and has been documented in many publications). A fact is not a theory. It remains unclear what the content (beyond the facts mentioned) of their theory is.

Response 2: Following your suggestions, we revised the following technical terms:

“Pattern” instead of “Theory”

“Crude AGE pattern” instead of “Crude AGEs theory”

“Diverse AGE pattern” instead of “Diverse AGEs theory”

“Multiple AGE pattern” instead of “Multiple AGEs theory”

Comment 3: The type 1 diverse AGEs theory appear to propose that one specific protein can be modified by different AGEs (section 6./ Figure 6). The authors then discuss the paper by Kinoshita et al., who specifically examined CMA modification of collagen and identified CMA pöeptides in glyoxal-modified type III collagen. The authors conclude that „Kinoshita et al. could not completely prove the type 1 diversity AGEs theory“, but that was not the aim of Kinoshita et al. and it is not clear why the experiments in Kinoshita et al. could have the potential to prove the „type 1 diversity AGEs theory“ (in the view of this reviewer the Kinoshita et al. experiments are not able to prove the theory, and were also not designed to do so).

Response 3: Following your suggestions, we removed the sentences and Figure 7 that detailed the information of CMA-modified peptides by Kinoshita et al from Section 6.1.

However, their investigation provides an important example of AGEs analysis using ESI-MS. Therefore, we retained the information in “Introduction” and Section 2.7. (Ref. 30).

Comment 4: The meaning of many sentences are not clear (likely gramatically not correct; though this reviewer don’t want to judge that, as he is not a native speaker): Only a few examples: line 339/340: „Because GC-MS can be used to

Response 4: We have deleted this sentence (Please see Response 3).

Comment 5: line 348: „We introduced one of these CMA-modified peptides (Figure 7); do the authors mean „We show one ......“?

Response 5: We have deleted this sentence (Please see Response 3).

Comment 6: line 351: „(one type of protein, but one molecular)“; meaning of „one molecular“ is unclear – is the expression incomplete?

Response 6: We have deleted this sentence (Please see Response 3).

Comment 7: line 462/463: „The procedure by which one type of AGE structure combines more than two proteins is difficult if researchers perform using MALDI-MS or ESI-MS analyses.“: does it refer to the „procedure“ by which AGEs combines more than two proteins or the procedure researcher are using to exmine this?

Response 7: Following your suggestions, we rewrote this sentence (yellow highlighted) in Section 7.2 as follows:

“MALDI-MS or ESI-MS analyses cannot accurately identify the one type of AGE structure that combines more than two proteins.”

Comment 8: line 464: „The peptide sequence without modifying the AGE structure ...“; does it mean „The peptide sequence without AGE-modification“?

Response 8: Following your suggestions, we rewrote this sentence (yellow highlighted) in Section 7.2 as follows:

“The peptide sequence without AGE-modification…”

Comment 9: Minor points: It would be helpful for the reader to list all abbreviations that are used more than once in the list at the end of the manuscript (e.g. 1,5-AF-AGEs, MGO-AGEs, TAGE).

Response 9: We rewrote the list of abbreviations.

Comment 10: Figure 7: the legend states that „One of the CMA-modified peptides ....“ is shown; but actually 3 peptides are shown in the Figure!

Response 10: We have deleted this sentence (Please see Response 3).

Comment 11: Figure 9, caption: should be „CEL-modified“ not „GEL-modified“

Response 11: We have included the new Figure 8, which was Figure 9 in the first submitted manuscript. “CEL-modified proteins” are described in Figure 8.

Round 2

Reviewer 2 Report

Comments and Suggestions for Authors

The authors addressed most, especially the important points and the manuscript could be published now in its present form.

Author Response

Dear Reviewer 2,

Thank you for review of our revised draft of my manuscript titled “Analysis of Crude, Diverse, and Multiple Advanced Glycation End-Product Patterns May be Important and Beneficial” to the journal Metabolites (manuscript ID: 2724292).

We are grateful to the Reviewer 2 for the thoughtful suggestions and insightful comments on our paper, which have enriched the manuscript and produced a better and more balanced account of the research.

In this stage, we inserted some sentences and the new references in Section 9 in our previous revised manuscript based on the Academic Editor’s Comments.

The inserted sentences were Yellow highlighted.

To describe our revised manuscript, we inserted six references (Ref. 105–108,111,112).

Ref. 105: PMID 23662155

Ref. 106: PMID 35417488

Ref. 107: PMID 23674882

Ref. 108: PMID 37005323

Ref. 111: PMID 37925002

Ref. 112: PMID 37832514
